# *Trypanosoma cruzi* DNA Identification in Breast Milk from Mexican Women with Chagas Disease

**DOI:** 10.3390/microorganisms12122660

**Published:** 2024-12-21

**Authors:** María del Pilar Crisóstomo-Vázquez, Griselda Rodríguez-Martínez, Verónica Jiménez-Rojas, Leticia Eligio-García, Alfonso Reyes-López, María Hernández-Ramírez, Francisco Hernández-Juárez, José Luis Romero-Zamora, Silvia Guadalupe Vivanco-Tellez, Fortino Solorzano-Santos, Victor M. Luna-Pineda, Guillermina Campos-Valdez

**Affiliations:** 1Laboratorio de Investigación en Parasitología, Hospital Infantil de México Federico Gómez, Ciudad de México 06720, Mexico; crisostomopilar@yahoo.com.mx; 2Laboratorio de Investigación en Patógenos Respiratorios y Producción de Biológicos, Hospital Infantil de México Federico Gómez, Ciudad de México 06720, Mexico; griseldargzmtz123@gmail.com (G.R.-M.); verozenemij@hotmail.com (V.J.-R.); leticiaeligio@yahoo.com.mx (L.E.-G.); 3Centro de Estudios Económicos y Sociales en Salud, Hospital Infantil de México Federico Gómez, Ciudad de México 06720, Mexico; alfonso.reyes.lopez@outlook.com; 4Hospital General de Santa María Huatulco, Santa María Huatulco 70980, Oaxaca, Mexico; mariara245@hotmail.com; 5Hospital Comunitario “Dr. Pedro Espinosa Rueda”, Pinotepa Nacional 71600, Oaxaca, Mexico; hjfranciscoooo@gmail.com; 6Departamento de Enfermedades Infecciosas, Hospital Infantil de México Federico Gómez, Ciudad de México 06720, Mexico; jlromeroz@yahoo.com; 7Unidad de Investigación en Enfermedades Infecciosas, Hospital Infantil de México Federico Gómez, Ciudad de México 06720, Mexico; sgvtellez02@gmail.com (S.G.V.-T.); solorzanof056@gmail.com (F.S.-S.)

**Keywords:** *Trypanosoma cruzi*, Chagas disease, breastfeeding, oral transmission, antibodies, vertical transmission route

## Abstract

(1) Background: Chagas disease is a public health problem affecting nearly 2 million women of reproductive age in Latin America. From these, 4–8% can transmit the infection to the foetus through the vertical route, whereas horizontal transmission through milk during breastfeeding remains controversial. Therefore, the presence of *Trypanosoma cruzi* (*T. cruzi*) DNA in the milk of women seropositive for Chagas disease was analysed to determine whether a relationship with the infection of their children can exist. (2) Methods: 260 pairs (mother–child) from four hospitals located in rural areas endemic to *T. cruzi* (state of Oaxaca) were studied. The presence of anti-*T. cruzi* antibodies in the serum of lactating women were determined by ELISA, whereas parasitic DNA in either breast milk or newborn’s blood was identified by PCR; (3) Results: The seroprevalence of infection in lactating women was 5.76%, and the frequency of infection detected by PCR in breast milk was 1.92%, while the frequency of infection in the blood of newborns was 1.92%. Pochutla-Oaxaca presented the highest number of positive cases in both breast milk and blood. The only risk factor found was the presence of the vector in the geographical area analysed, favouring the parasite’s transmission. Overall, the results suggest a probable transmission of *T. cruzi*, although whether it was through breastfeeding or through the blood during delivery could not be determined. (4) Conclusions: *T. cruzi* DNA was identified in lactating women’s milk and newborn blood, which is probable evidence of transmission through breastfeeding; nevertheless, future studies must be performed to confirm the presence of the parasite, alive or dead.

## 1. Introduction

Chagas disease is caused by the hemoflagellate protozoan parasite *Trypanosoma cruzi* (*T. cruzi*). The parasite is vectorially transmitted to humans by the hematophagous insect of the generous Triatoma [1]. This neglected tropical disease affects about 8 million people in Latin America and causes losses of US$1.2 billion annually [2,3]. Due to the constant population migration from Central America, South America, and Mexico endemic areas, Chagas disease is also found in non-endemic countries such as the United States, Spain, and Portugal. For instance, Gascon et al. reported that among the immigrant population in Spain, Bolivian immigrants showed the highest *T. cruzi* prevalence (6.75%) [4]. Thus, the appearance of the disease has been reported to be due to a non-vector transmission (congenital or blood transmission) from infected immigrants [4], generating concern in developing countries with higher migration rates.

Mexico is one of the main endemic countries, possessing 39 documented Triatoma species, with at least 21 of them reported to be infected by *T. cruzi* [5]. Since two-thirds of the Mexican territory is considered at risk for transmission through the vector, the number of cases of Chagas disease is estimated to be 1.1 million people infected in Mexico annually, with an incidence of 0.80 per 100,000 population [6,7]. The states of Veracruz, Chiapas, Quintana Roo, Oaxaca, Morelos and Yucatán have recorded the highest incidence rates in Mexico [8,9].

As *T. cruzi* is carried in the faeces of infected triatomine, infection occurs when the vector bites an exposed area of human skin or mucous membranes, and immediately after feeding, it defecates near the bite, allowing the passage of the parasite. Thus, the parasite enters the bloodstream through the skin, facilitated by self-inflicted scratching and proteolytic enzymes in the triatomine’s saliva [10]. Moreover, non-vectorial transmission routes in the infected population include blood transfusion, vertical transmission from an infected pregnant woman to her foetus, organ transplants, and the ingestion of food contaminated with the parasite [11]. In infected pregnant women, the parasite can infect the foetus by crossing the placental barrier, resulting in congenital Chagas disease in approximately 20–30% of infected infants [12]. Therefore, it has been proposed both prevention in girls, female adolescents, and women of fertile age and antenatal screening for infection in pregnant women are efficient preventive measures in controlling congenital transmission of *T. cruzi* [13].

The World Health Organization (WHO) recommends conventional serological tests, such as indirect hemagglutination assay, indirect immunofluorescence assay, or Enzyme-Linked ImmunoSorbent Assay (ELISA), based on crude or recombinant antigens as diagnostic tests in Chagas disease [14]. In Mexico, the Chagas prevalence in pregnant women is 3%, whereas the vertical transmission rate is 8% [15]. Detection of antibodies (Ab) against *T. cruzi* in serum is currently used to diagnose Chagas disease in pregnant women. In contrast, determining the presence of parasites in blood by polymerase chain reaction (PCR) is recommended in newborns [14]. Hence, several studies in Latin America have focused on determining the seroprevalence in pregnant women and congenital transmission of *T. cruzi* [16,17,18]. In a previous prospective study of a transversal cohort of samples obtained from Oaxaca, Mexico City, and Jalisco, a seroprevalence of infection in pregnant women of 4.12–12.02% was found, whereas the congenital transmission rate was 1.5–9.1% [17].

The data regarding the horizontal transmission route of *T. cruzi* infection are controversial. For instance, in previous works, *T. cruzi* trypomastigotes were localised in milk from female mice without blood contamination [19]. Still, in humans, the transmission of trypomastigotes in infected mothers’ milk to newborns through lactation was conflicting as the authors emphasised that the collected milk was contaminated by blood and, therefore, could be responsible for the horizontal transmission route through milk [20,21]. On the other hand, human milk samples infected with *T. cruzi* trypomastigotes were able to generate infection when orally inoculated to mice [22].

Despite oral transmission through human milk contaminated with trypomastigotes is possible, natural transmission through breastfeeding has not been fully demonstrated in Chagas disease [23]. Since breast milk is the newborns’ only food source containing essential nutritional and immunological development elements during the first year of life, this study aimed to determine whether the presence of *T. cruzi* in the milk of lactating women seropositive for Chagas disease could be responsible for the infection in newborns.

## 2. Materials and Methods

### 2.1. Study Population and Ethical Statement

The state of Oaxaca, Mexico, is an endemic area to the *Rhodnius prolixus*, *Triatoma barberi*, and *T. bolívar* vectors. This study was conducted from October 2019 to November 2021. It included mother–child pairs in the breastfeeding stage from four hospitals in Oaxaca: The Hospital General (HG) de San Pedro Pochutla, Centro de Salud con Servicios Ampliados (CESSA) in Santa Cruz Huatulco, Hospital Comunitario (HC) de Santa María Huatulco, and HC “Dr. Pedro Espinosa Rueda” in Santiago Pinotepa Nacional (Appendix A). The sample size was calculated considering the number of births attended per year in each hospital and the reported prevalence of the vertical Chagas transmission route [17].

The women were given a questionnaire to obtain general data: name, age, address, place of birth and origin. Moreover, to determine possible risk factors, the following epidemiological and obstetric information was considered: knowledge of the transmitting vector, presence of the transmitting vector in their home, number of blood transfusions received, number of pregnancies, resolution of pregnancies and weeks of gestation by date of last menstrual period. The newborn data included the anthropometric data: date of birth, sex, age (determined by the Capurro method), pregnancy resolution, the Apgar score (colour, heart rate, reflexes, muscle tone, and respiration), somatometry data (weight, height and head circumference), and presence of liver, heart and/or digestive disorders.

For breast milk sample obtention, 260 women were asked to provide approximately 5 mL of breast milk. Each breast milk sample was mixed with the nucleic acid preservation reagent (DNA/RNA Shield, Zymo Research, Irvine, CA, USA) to transport the sample at room temperature to avoid nucleic acid damage, degradation or contamination. In addition, 5 mL of blood was taken by venipuncture and placed in a vacutainer tube (Becton Dickinson, Franklin Lakes, NJ, USA) to obtain serum. Regarding 261 newborns (one twin), a drop of blood was taken by heel prick and put on sterile filter paper. As negative controls, 30 breastfeeding women seronegative to *T. cruzi* and their newborns were also included.

The project was reviewed and approved by the Research Committee, Ethics Committee, and Biosecurity Committee from HIMFG with grant number HIM/2018/095 SSA 1533. The work was carried out following the international guidelines of the Helsinki postulates. Blood samples were collected according to the ethics and biosafety protocols published in the laboratory’s standardised guidelines. Since the general health law regulations on research state that sampling by either middle finger puncture or venipuncture is considered a minimal-risk procedure, informed consent was obtained from each participant.

### 2.2. Anti-T. cruzi Antibodies by Indirect Enzyme-Linked Immunosorbent Assay (ELISA)

The Chagas IgG ELISA kit with total *T. cruzi* antigen (DGR ELISA; 96% sensibility and 97% specificity) and the recombinant antigen MicroELISA (ACCUTRACK CHAGAS; 99.5% sensibility and 99.2% specificity) were performed to evaluate anti-*T. cruzi* IgG in the serum of lactating women. The procedures were performed following the specifications of the suppliers. A positive test was considered when the optical density of the analysed samples was more significant than the value of the cohort points for each kit, with optical density (OD) values higher than 0.167 and 0.162, respectively [24].

### 2.3. T. cruzi-Specific GAPDH Gene by Polymerase Chain Reaction (PCR)

*T. cruzi* in breast milk and newborn blood was identified using polymerase chain reaction (PCR). Briefly, DNA was extracted from 1 mL of breast milk using the Food DNA Isolation Kit (Norgen, Thorold, ON, Canada), while the DNA from the drop of whole blood on filter paper was extracted using the Quick-DNA Miniprep Plus Kit (Zymo Research), following the recommendations of both suppliers.

Since GAPDH is an essential enzyme in the glycolytic pathway that controls the parasite’s energy production and is a target of antiparasitic compounds, the amplification primer sequence was obtained from Arrollo-Olarte et al. with the following sequences: GAPDH-F (5′-AGCATACAGGAGATCGACGC-3′) and GAPDH-R (5′-CGTAAATGGAGCTGCGGTTG-3′) [25]. The reaction mixture was prepared with 12.5 μL of Dream Taq Green PCR master mix (2×), 1 μL of the forward primer (50 nM), 1 μL of the reverse primer (50 nM), 5 μL of DNA (100 ng) and at to 25 μL of nuclease-free water as final volume. The amplification program included (1) one cycle of denaturation at 95 °C for 3 min, (2) 35 cycles of denaturation, annealing and extension at 95 °C for 30 s, 58° for 30 s and 72° for 30 s, (3) a final cycle at 72 °C for one minute. This PCR reaction was performed in an Invitrogen MiniAmp thermocycler. The controls included *T. cruzi* strain Tulahuen (ATCC 30266) as positive control and confirmed human DNA without Chagas disease as negative control, whereas PCR mix without DNA template was a reaction control. The amplificons were run by electrophoresis in a 1% agarose gel, stained with SYBR safe (DNA gel stain from Invitrogen) and visualised on iBright Imaging System (CL 1000, Invitrogen, Waltham, MA, USA).

### 2.4. Statistical Analysis

The ELISA and PCR results and clinical data analysis were used to calculate the relative and absolute frequencies from all the categorical variables, including all variables’ absolute and relative frequencies since they were categorical. The association between the variables and the results obtained was evaluated using tests of differences in proportions for qualitative data and medians for quantitative data with Pearson’s χ^2^ and Fisher exact tests, using an α level of 5% as a cut-off point to determine statistically significant results (*p* < 0.05). All procedures were analysed with the STATA statistical package, version 18.

## 3. Results

Two hundred and sixty mother–child pairs from the state of Oaxaca participated in this study. From these, 70 (26.92%) were from the CESSA, 70 (26.92%) from the HG-San Pedro Pochutla, and 69 (26.4%) from the HG-Dr. Pedro Espinosa Rueda, and 51 (19.62%) from the HC-Santa María Huatulco. The average age of the participating women was 24.76 ± 6.49 years. Of the total women, 36.15% were primigravida, 63.85% were multigravida, 50.76% had a caesarean, 12.68% had a miscarriage, and none had any death birth. Regarding characteristics of the newborns, 47.69% were female, the mean gestational age was 38.84 ± 2.18 weeks, the mean APGAR at 1 min was 8.13 ± 1.31, and the APGAR at 5 min was 8.96 ± 0.45, the weight was 3111.24 ± 548.91 g, the size was 49.32 ± 3.13 cm, and the head circumference was 34.71 ± 3.60 cm.

### 3.1. Positivity Frequency from Blood PCR and Maternal Seropositivity

From the total mother–child pairs (each mother with her respective child; N = 260), only 19 pairs (7.3%) were positive by ELISA, PCR or both tests and considered positive for either Chagas infection or the presence of *T. cruzi* DNA. Of these, six (31.59%) were from the HG-San Pedro Pochutla and five (26.31%) from the HG-Dr Pedro Espinosa Rueda, five (26.31%) were from the CESSA, three (15.79%) were from the HC-Santa María Huatulco. On the other hand, only fifteen women (78.94%) were positive for anti-*T. cruzi* IgG antibodies and five positives (26.31%) in breast milk for PCR, who were asymptomatic to Chagas disease. From the mothers with *T. cruzi*-positive breast milk, their newborns (n = 5) were identified with *T. cruzi* DNA by blood PCR. Interestingly, only one mother–child pair was positive for both ELISA and PCR tests (Table 1). Regarding the results obtained with the PCR test in milk from lactating women and in blood from newborns from the four hospitals studied, a highly significant association (*p* < 0.005) was observed between PCR positivity in milk and PCR positivity in newborns’ blood.

### 3.2. Relationship Between Women with Chagas Disease and Clinical Obstetric Data

The women with Chagas disease showed no association between infection with *T. cruzi* and the parameters studied as indicators of vertical transmission: primiparous women, multiple pregnancies, caesarean births, abortions, premature births (weeks of gestation) and the hospital of origin. Regarding the age of the women, a higher proportion was found in the groups 18 to 24 and 25 to 34 years old in women with positive serology for *T. cruzi* (40 to 50%). Interestingly, a significant association was observed in women with ≥25 years from HG-San Pedro Pochutla (Table 2).

Noteworthily, the prevalence of maternal positivity to PCR was 1.92%, and the hospital with the highest number of positive cases in women and children was HG-San Pedro Pochutla (2.89%). However, no statistical significance was obtained between the results of the PCR tests in breast milk and the clinical obstetric data in the four hospitals (Table 3).

### 3.3. Relationship Between Newborns with Chagas Disease and Somatometry Data

Infants born from women with positive serology and PCR for *T. cruzi* did not show an association between infection and the parameters considered indicators of infection acquired through the vertical route. Anthropometric data at birth of infants positive for *T. cruzi* by PCR was within normal values (Table 4). Since no associations between PCR-positive and PCR-negative infants were observed, it was suggested that infants with suspected *T. cruzi* infection were asymptomatic. Therefore, based on the laboratory results, it can be suggested that there was only suspicion of infection.

Of the fifteen women with a positive ELISA test, only one was positive for PCR and the PCR of her child (Table 1). This case is the most interesting to our purpose because it is likely that the infant could have been infected by its mother. Still, whether this infection occurred by breast milk or the vertical transmission route cannot be determined. Despite the PCR test in milk identifying the *T. cruzi* DNA, it is impossible to determine whether the parasite is alive or dead with this test. Thus, the determination of *T. cruzi* DNA in the newborns will be essential to demonstrate the presence of the infection in this mother–child pair.

## 4. Discussion

Breastfeeding as a transmission mechanism for *T. cruzi* infection has not been well described. Although several works describing the presence of the parasite in breast milk have been reported, some authors report their absence, leading to the conclusion that this transmission route remains controversial [23]. The first studies on the transmission of Chagas disease through breastfeeding were observational and were carried out on experimentally infected mice and rats [26]. Information regarding *T. cruzi* transmission through breastfeeding is scarce in humans and presents contrasting results. For example, in 1936, Mazza and collaborators carried out the first human studies, finding trypomastigotes in the milk of women in the acute phase of Chagas disease. However, it could not be demonstrated that the transmission was through breastfeeding since the milk of some women was contaminated with blood from cracked nipples [27]. In contrast, in a work by Bittencourt et al. conducted on 78 Chagasic mothers, no *T. cruzi* parasite in colostrum or milk could be found in either the women or the nursing children [27]. The results obtained in this study demonstrated the presence of the parasite’s DNA in the milk of women who were reactive to serological tests and the presence of DNA in the blood of their nursing children; however, it was not possible to identify the transmission route, whether it was through breastfeeding, through infection with blood from the nipple or during childbirth. Animal models could help demonstrate breastfeeding as a transmission route for *T. cruzi* infection.

The state of Oaxaca is one of the Mexican regions with the highest risk of contracting *T. cruzi* [8]. In this study, we observed an increase in the number of cases in San Pedro Pochutla, a region located north of Oaxaca; since this region lacks control programs for triatomine bugs, the vector transmission is likely active in this geographic area, contributing to the dissemination of the parasite. The high prevalence of lactating women infected with *T. cruzi* highlights the need to implement diagnostic strategies to identify asymptomatic infected women attending hospitals or maternity wards, and thus can be treated during and after their pregnancies to avoid transmission to their children. Interestingly, a case of a mother–child pair, both positive by PCR and serological tests, was identified in the Pochutla hospital. In this case, transmission through breastfeeding could be suggested; however, as the sample could not be used to either isolate the parasite in culture media or to observe its presence by microscopy, no definitive conclusion can be made. Another plausible explanation is that the DNA present in the blood of the newborns is due to the opsonising capacity of the mother’s anti-*T. cruzi* antibodies. Carlier et al. reported that infected pregnant women can transmit the antibodies against the parasite to the foetus. In turn, the antibodies, when opsonising parasites, can promote phagocytosis and thus the destruction of *T. cruzi* by cells expressing FcγR provided that these cells are sufficiently activated in an inflammatory context, as opposed to what occurs in infected pregnant women who transmit the parasite to their newborns through the vertical route [28].

The presence of *T. cruzi* DNA in newborns from infected women was identified in this work. Unfortunately, the transmission through the ingestion of milk from infected women could not be demonstrated since the presence of the live parasite was not identified. A work by Perinetti et al. showed that in female mice infected with trypomastigotes of the Y strain of *T. cruzi*, only a few trypomastigotes were ingested by the offspring. The low trypomastigotes ingested with the mother’s antibodies were responsible for the parasite’s lack of transmission to the mouse pups [19]. The work results showed that the women who were reactive by serology were 20 to 30 years old, and all were asymptomatic. These data are in agreement with recent works reporting that the age with the highest prevalence of Chagas disease in women is 20 years, mainly in areas of high endemicity. This finding highlights the need to carry out parasitological and serological monitoring of women infected with the parasite, especially when pregnant, to carry out effective strategies to prevent transmission before and after delivery to the offspring and during lactation. Identifying the mother–child binomial infected with *T. cruzi* is essential for developing strategies that prevent transmission to the infant without affecting their development and feeding.

## 5. Conclusions

In this study, *T. cruzi* DNA was identified in lactating women’s milk, but only in one of these mother–child pairs did the mother have anti-*T. cruzi* IgG antibodies. Her corresponding newborn was also positive for *T. cruzi* DNA, suggesting circulating parasites could be present in colostrum. Nevertheless, several mothers had anti-*T. cruzi* IgG antibodies, but *T. cruzi* DNA was not identified in breast milk and newborns’ blood; probably, the antibodies are immunological memory, or very few parasites were circulating in their blood. Future studies must be performed to confirm the presence of the parasite, alive or dead.

## Figures and Tables

**Table 1 microorganisms-12-02660-t001:** Number of patients positive for PCR or ELISA tests.

	CESSA					Hospital Santa Maria Huatulco			Hospital de San Pedro Pochutla						Hospital General “Dr. Pedro Espinosa Rueda”				
	P1	P2	P3	P4	P5	P6	P7	P8	P9	P10	P11	P12	P13	P14	P15	P16	P17	P18	P19
AGE	18	17	18	41	24	16	34	25	27	43	41	19	28	23	36	18	27	22	28
GESTATIONS	2	1	1	5	2	1	3	2	1	8	2	3	4	1	3	1	1	1	2
ABORTIONS	0	0	0	2	0	0	0	0	0	0	1	0	0	0	0	0	0	0	0
ELISA	−	+	+	+	+	−	+	+	+	+	+	−	+	+	+	+	+	−	+
PCR IN BREAST MILK	+	−	−	−	−	+	−	−	−	−	+	+	−	−	−	−	−	+	−
PCR IN NEWBORN’S BLOOD	+	−	−	−	−	+	−	−	−	−	+	+	−	−	−	−	−	+	−

CESSA, Centro de Salud con Servicios Ampliados; ELISA, Enzyme-Linked ImmunoSorbent Assay; PCR, polymerase chain reaction.

**Table 2 microorganisms-12-02660-t002:** Relationship between positive maternal serology and clinical obstetric data.

	CESSA	Hospital Santa Maria Huatulco	Hospital de San Pedro Pochutla	Hospital General “Dr. Pedro Espinosa Rueda”
AGE (YEARS)	*p* = 0.140	*p* = 1.000	*p* = 0.034	*p* = 0.455
<18	25% (1)	0 (0)	0 (0)	0 (0)
18–24	50% (2)	50% (1)	20% (1)	25% (1)
25–34	0 (0)	50% (1)	40% (2)	50% (2)
35–44	25% (1)	0 (0)	40% (2)	25% (1)
PRIMIGRAVIDA	*p* = 0.593	*p* = 0.534	*p* = 1.000	*p* = 1.000
(n/N = %)	50% (2/4)	0	40% (2/5)	50% (2/4)
MULTIGRAVIDA	*p* = 0.593	*p* = 0.534	*p* = 1.000	*p* = 1.000
(n/N = %)	50% (2/4)	100% (2/2)	60% (3/5)	50% (2/4)
CESAREAN	*p* = 1.000	*p* = 0.235	*p* = 1.000	*p* = 0.33
(n/N = %)	50% (2/4)	100% (2/2)	60% (3/5)	25% (1/4)
GESTATION WEEKS	*p* = 0.385	*p* = 1.000	*p* = 1.000	*p* = 0.484
<38	25% (1/4)	0	0	0
38 to 40	75% (3/4)	100% (2/2)	100% (5/5)	75% (3/4)
>40	0	0	0	25% (1/4)

**Table 3 microorganisms-12-02660-t003:** Relationship between positive PCR in breast milk and clinical obstetric data.

	CESSA	Hospital Santa Maria Huatulco	Hospital de San Pedro Pochutla	Hospital General “Dr. Pedro Espinosa Rueda”
AGE (YEARS)	*p* = 1.000	*p* = 0.157	*p* = 0.127	*p* = 1.000
<18	0 (0)	100% (1)	0 (0)	0 (0)
18–24	100% (1)	0 (0)	50% (1)	100% (1)
25–34	0 (0)	0 (0)	0 (0)	0 (0)
35–44	0 (0)	0 (0)	50% (1)	0 (0)
PRIMIGRAVIDA	*p* = 1.000	*p* = 0.353	*p* = 0.534	*p* = 0.406
(n/N = %)	0	100% (1/1)	0	100% (1/1)
MULTIGRAVIDA	*p* = 1.000	*p* = 0.353	*p* = 0.534	*p* = 0.406
(n/N = %)	100% (1/1)	0	100% (2/2)	0
CESAREAN	*p* = 1.000	*p* = 0.490	*p* = 1.000	*p* = 0.464
(n/N = %)	0	100% (1/1)	50% (1/2)	0
GESTATION WEEKS	*p* = 1.000	*p* = 0.137	*p* = 1.000	*p* = 1.000
<38	0	0	0	0
38 to 40	100% (1/1)	0	100% (2/2)	100% (1/1)
>40	0	100% (1/1)	0	0

**Table 4 microorganisms-12-02660-t004:** Relationship between positive PCR in newborns and somatometry data.

	CESSA	Hospital Santa Maria Huatulco	Hospital de San Pedro Pochutla	Hospital General “Dr. Pedro Espinosa Rueda”
GESTATIONAL AGE (weeks)	*p* = 0.543	*p* = 0.451	*p* = 0.033	*p* = 1.000
<38	100% (1/1)	100% (1/1)	50% (1/2)	0
38 to 40	0	0	0	100% (1/1)
>40	0	0	0	0
FEMALE	*p* = 1.000	*p* = 1.000	*p* = 0.496	*p* = 0.435
(n/N = %)	100% (1/1)	0 (0/1)	100% (2/2)	100% (1/1)
DELIVERY 1	*p* = 1.000	*p* = 0.451	*p* = 1.000	*p* = 1.000
(n/N = %)	100% (1/1)	0 (0/1)	50% (1/2)	100% (1/1)
CESAREAN	*p* = 1.000	*p* = 0.451	*p* = 1.000	*p* = 1.000
(n/N = %)	0 (0/1)	100% (1/0)	50% (1/2)	0 (0/1)
WEIGHT (grams)	*p* = 1.00	*p* = 1.000	*p* = 1.000	*p* = 1.000
Normal: 2.5–4.0	100% (1/1)	100% (1/1)	100% (2/2)	100% (1/1)
Low: <2.5	0	0	0	0
High: >4.0	0	0	0	0
SIZE (cm)	*p* = 1.000	*p* = 1.000	*p* = 1.000	*p* = 1.000
Normal: 45–53	100% (1/1)	100% (1/1)	100% (2/2)	100% (1/1)
Low: <45	0	0	0	0
High: >53	0	0	0	0
HEAD CIRCUMFERENCE (cm)	*p* = 1.000	*p* = 1.000	*p* = 0.513	*p* = 1.00
Normal: 33–35	100% (1/1)	100% (1/1)	50% (1/2)	100% (1/1)
Low: <33	0	0	50% (1/2)	0
High: >35	0	0	0	0
APGAR (1 min) PCR + Mean (SD)	9	8	7.5 (0.7)	8
APGAR (5 min) PCR + Mean (SD)	9	9	8.5 (0.7)	8

## Data Availability

The corresponding author (G.C.-V.) can provide informed consent and clinical demographic datasets from the population upon reasonable request. All data were anonymised before analysis.

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
