# Peer review of "Trypanosoma cruzi DNA Identification in Breast Milk from Mexican Women with Chagas Disease"

_microorganisms, 2024, doi:10.3390/microorganisms12122660_

Round 1
Reviewer 1 Report
Comments and Suggestions for Authors
I believe that the topic addressed by the authors of the article entitled “Trypanosoma cruzi DNA identification in breast milk from Mexican women with Chagas disease” is important and topical, as it concerns the health care of children and mothers in general and Chagas disease in particular, which represents a serious health and social problem for endemic countries. In this regard, I would like to congratulate the authors on the ambitious goal they have set themselves to elucidate a possible mechanism of infection in this serious disease. However, I have some comments that I would like to share with the authors.
Abstract - I have no comments
Introduction - lines 48-51 The authors need to clarify whether cases in the USA, Spain and Portugal were diagnosed in migrants or natives who had travelled to endemic areas.
Materials and Methods
line 136 - Move sentence „ The corresponding author (G.C.) can provide informed consent upon request“ to Data Availability Statement
line 144 - It would be useful if the authors provided sensitivity/specificity parameters of the diagnostic kit.
Results
line 187 - The sentence "Of the total mother-child pairs, only 19 women and newborns presented with Chagas (7.3%)" is unclear. What exactly do the authors mean? What did the pairs present - disease, laboratory evidence of infection, or something else? The sentence needs to be rewritten.
In my opinion, Table 1 should be reformatted as it does not present the data well in its current form. The authors should consider whether swapping the column and row positions would produce a better result.
line 187 - I was really puzzled by the first age group defined by the authors (10 - 20 years) when it comes to pregnant and breastfeeding women. Was there a pregnant and then breastfeeding 10 year old? I recommend that the authors look at the ESOMAR guidelines on demographics and age groups in surveys (https://esomar.org/uploads/attachments/cl5v19rsk1f9rew3vhglmzn55-esomar-demographics-best-practice-recommendation-on-age-for-consultation.pdf). There they are classified in a way that I think is more appropriate, which is as follows:
Under 18 years
â–ª 18-24 years
â–ª 25-34 years
â–ª 35-44 years
â–ª 45-54 years, etc.
I think the authors should revise the age groups and perform a new statistical analysis accordingly.
lines 219-220 - The authors suggested that the lack of differences in somatometric data between PCR positive and negative infants might suggest that potentially infected infants are asymptomatic. What is the reality? Do the authors have information on the objective status of these infants at the time of the study? Did the infants have any symptoms indicating a disease state or were they objectively clinically healthy? If there were no symptoms, it is more accurate to say that there was only a suspicion of infection based on the laboratory results.
lines 228-229 - Another puzzling opinion. How would serological testing be essential to prove infection in neonates? It is well known that IgG antibodies can cross the placental barrier and enter the fetal circulation, where they circulate for some time after birth. Maybe they will look for seroconversion of specific antibodies by follow-up serological screening? The authors make no mention of testing for markers such as specific IgM antibodies or IgG avidity that might indicate recent infection. In any case, this statement is very loud and not quite true.
Conclusions
The conclusion repeats some of the data presented in the results and discussion sections. My suggestion is to include only the most important findings of the study in this section.
Author Response
Reviewer 1
I believe that the topic addressed by the authors of the article entitled “Trypanosoma cruzi DNA identification in breast milk from Mexican women with Chagas disease” is important and topical, as it concerns the health care of children and mothers in general and Chagas disease in particular, which represents a serious health and social problem for endemic countries. In this regard, I would like to congratulate the authors on the ambitious goal they have set themselves to elucidate a possible mechanism of infection in this serious disease. However, I have some comments that I would like to share with the authors.
We really appreciate the time spent thoroughly reviewing our work. We strongly believe all the comments and suggestions made to the manuscript were valuable and appropriate. We accepted all the comments/ suggestions, and changes were made to address every concern successfully. A point-by-point report is provided below. We hope all the changes made to the manuscript will be sufficient to make the work more precise and readable.
Noteworthy, all the changes made to the manuscript were highlighted in red for better visualisation.
Comment 1. Abstract - I have no comments
Response 1. No apply
Comment 2. Introduction - lines 48-51 The authors need to clarify whether cases in the USA, Spain and Portugal were diagnosed in migrants or natives who had travelled to endemic areas.
Response 2. We really appreciate the observation. To make this sentence clearer, it was changed from: “the cases of Chagas disease reported in the U.S., Spain, and Portugal are primarily linked to migration from Latin America, where the disease is endemic.” to “For instance, Gascon et al., reported that among the immigrant population in Spain, Bolivian immigrants showed the higher T. cruzi prevalence (6.75%) [4]. Thus, the appearance of the disease has been reported to be a non-vector transmission (congenital or blood transmission) from infected immigrants, generating concern in countries with higher migration rates. (lines 54-58)
Materials and Methods
Comment 3. line 136 - Move sentence „ The corresponding author (G.C.) can provide informed consent upon request“ to Data Availability Statement
Response 3. We agree with the reviewer´s suggestion. Hence, the sentence “The corresponding author (G.C.) can provide informed consent and clinical-demographic datasets from the population upon request. All data were anonymised before analysis.” was moved to the section “Data Availability Statement” (lines 324-326).
Comment 4. line 144 - It would be useful if the authors provided sensitivity/specificity parameters of the diagnostic kit.
Response 4. Following the reviewer´s comment, the sentence was modified to include the sensitivity/specificity of the diagnostic kit. This sentence now reads as: “The Chagas IgG ELISA with total T. cruzi antigen (DGR ELISA; 96% sensibility and 97% specificity) and the recombinant antigen MicroELISA (ACCUTRACK CHAGAS; 99.5% sensibility and 99.2% specificity) were performed to evaluate anti-T. cruzi Ab in the serum of lactating women” (lines 145-146).
Results
Comment 5. line 187 - The sentence "Of the total mother-child pairs, only 19 women and newborns presented with Chagas (7.3%)" is unclear. What exactly do the authors mean? What did the pairs present - disease, laboratory evidence of infection, or something else? The sentence needs to be rewritten.
Response 5. This concern is quite pertinent. To make the sentence more precise, it was rewritten as “From the total mother-child pairs (each mother with her respective child; N = 260), only 19 pairs (7.3%) were positive by ELISA, PCR or both tests and considered positive for either Chagas infection or the presence of T. cruzi DNA” (lines 195-203)
Comment 6. In my opinion, Table 1 should be reformatted as it does not present the data well in its current form. The authors should consider whether swapping the column and row positions would produce a better result.
Response 6. To address this concern, Table 1 was reformatted to represent the data better (Page 5).
Comment 7. line 187 - I was really puzzled by the first age group defined by the authors (10 - 20 years) when it comes to pregnant and breastfeeding women. Was there a pregnant and then breastfeeding 10 year old? I recommend that the authors look at the ESOMAR guidelines on demographics and age groups in surveys (https://esomar.org/uploads/attachments/cl5v19rsk1f9rew3vhglmzn55-esomar-demographics-best-practice-recommendation-on-age-for-consultation.pdf). There they are classified in a way that I think is more appropriate, which is as follows:
Under 18 years
â–ª 18-24 years
â–ª 25-34 years
â–ª 35-44 years
â–ª 45-54 years, etc.
I think the authors should revise the age groups and perform a new statistical analysis accordingly.
Response 7. We agree with the reviewer´s comment. A pregnant 10-year-old girl was included in this study; however, she was negative for T. cruzi in both PCR and ELISA tests. For this reason, she was excluded from Tables 2 and 3. We really appreciate the suggestion of using the ESOMAR guidelines. Following this advice, a new classification of women´s age was made, and a new statistical analysis was performed in Tables 2 and 3.
Comment 8. lines 219-220 - The authors suggested that the lack of differences in somatometric data between PCR positive and negative infants might suggest that potentially infected infants are asymptomatic. What is the reality? Do the authors have information on the objective status of these infants at the time of the study? Did the infants have any symptoms indicating a disease state or were they objectively clinically healthy? If there were no symptoms, it is more accurate to say that there was only a suspicion of infection based on the laboratory results.
Response 8. We agree with the reviewer; the sentence was rewritten to avoid misunderstandings. Now it reads: “Anthropometric data at the birth of infants positive to T. cruzi by PCR was within normal values (Table 4). Since no associations between PCR-positive and PCR-negative infants were observed, it was suggested that infants with suspected T. cruzi infection were asymptomatic. Therefore, based on the laboratory results, it can be suggested that there was only suspicion of infection.” (lines 232-236)
Comment 9. lines 228-229 - Another puzzling opinion. How would serological testing be essential to prove infection in neonates? It is well known that IgG antibodies can cross the placental barrier and enter the fetal circulation, where they circulate for some time after birth. Maybe they will look for seroconversion of specific antibodies by follow-up serological screening? The authors make no mention of testing for markers such as specific IgM antibodies or IgG avidity that might indicate recent infection. In any case, this statement is very loud and not quite true.
Response 9. Following the reviewer´s comment, the line was rewritten to make it more precise: “The PCR test in milk identified the T. cruzi DNA; however, it is impossible to determine whether the parasite is alive or dead. Thus, the determination of T. cruzi DNA in the newborn will be essential to demonstrate the presence of the infection in this mother-child pair.” (lines 243-247)
Conclusions
Comment 10. The conclusion repeats some of the data presented in the results and discussion sections. My suggestion is to include only the most important findings of the study in this section.
Response 10. Following the reviewer´s suggestion, the “Conclusions” section was modified in the manuscript to be more concise.
Reviewer 2 Report
Comments and Suggestions for Authors
Chagas disease (CD) is a neglected tropical disease with significant morbidity and mortality.
It presents clinically in two distinct phases, acute (apparent or not) and chronic, the latter of which can manifest in undetermined, cardiac, digestive or cardiodigestive forms. The cardiac form is responsible for the high morbidity and mortality of Chagas disease, and the clinical manifestations of chronic Chagas heart disease (CCD) are grouped into three syndromes: arrhythmic, heart failure and thromboembolic.
Despite the significant reduction in the incidence of cases of acute Chagas disease (ACD), the systematic occurrence of these cases related to oral transmission through the ingestion of contaminated food, mainly in the Amazon region, as well as to extra-household vector transmission, with accidental exposure to the sylvatic cycle of the etiological agent, has become evident in the last 15 years.
This manuscript submitted by María del Pilar Crisóstomo-Vázquez and collaborators demonstrated that the presence of IgG anti-T. cruzi in the serum of lactating women was identified using the ELISA test from four hospitals in the state of Oaxaca. These hospitals are located in geographic areas at high risk of T. cruzi infection due to the presence of the vector in these regions. The PCR test identified the presence of T. cruzi DNA in the breast milk of lactating women who attended these four hospitals in the endemic areas of Oaxaca/Mexico. The T. cruzi DNA obtained from the blood of the infants included in the study was amplified by PCR. The only conclusive risk factor found in this study was the presence of the vector, which favors infection by the parasite. Although the results suggest a probable transmission of T. cruzi, the route could not be determined. All infants with positive results should be monitored for medical supervision and rule out the future development of megacardiopathy associated with Chagas disease. The use of murine models may help determine whether breastfeeding can cause Chagas disease. This manuscript is interesting, mainly due to the increased interest in infection via the oral route and the little information that exists about possible infection through breastfeeding.
Author Response
Response 1. We really appreciate the interest and feedback given to our work. Following the reviewer´s suggestion, a prospective study will be considered with ELISA and PCR-positive mother-child pairs, but with a main focus on mega cardiopathy development associated with Chagas disease. The newly submitted version of the manuscript addresses all the concerns raised, and changes were made in Table 2 to better represent the results.
We strongly believe that these modifications have improved the manuscript, making it more readable and understandable in this version.
To note, all the changes made to the manuscript were highlighted in red for better visualization.
Round 2
Reviewer 1 Report
Comments and Suggestions for Authors
I believe that the text of the article, after the corrections, is much better than the original. I have no substantive comments on the revised text, only a suggestion for two minor corrections:
Page 8, line 272 - I suggest replacing the term "incidence" with "prevalence" because evidence of infection in the absence of clinical symptoms may raise suspicion of disease but does not prove disease.
Page 9, lines 311-312 - I would suggest that the authors remove the sentence "In addition, age was the only demographic factor that had an association with mother-child pairs". I think it is redundant and does not contribute significantly to the conclusion.
Author Response
Again, we appreciate the reviewer's suggestions, which have been resolved. We consider that we have met the requested requirements. The new version of the manuscript was submitted as V3.